



# Improving Risk Assessment of Subaqueous Landslides and Tsunami Generation by Understanding the Roles of Pore Pressure and Sediment Cohesion

Maxwell M. W. Silver[1,2], Elizabeth Reddy[3], Brandon Dugan[1, 4]

[1]Hydrologic Science and Engineering Program, Colorado School of Mines, Golden, CO, 80401, USA
[2]Institute de Physique du Globe Paris, Université Paris Cité Sismologie (current address), Paris, 75013, France
[3]Department of Engineering, Design, & Society, Colorado School of Mines, Golden, CO, 80401, USA
[4]Department of Geophysics, Colorado School of Mines, Golden, CO, 80401, USA

*Correspondence to*: Brandon Dugan (dugan@mines.edu)

**Abstract.** Subaqueous landslides present a hazard to communities through their ability to damage infrastructure and generate tsunamis. Efforts to assess tsunami risk often depend on observations from prior events, such as volume of sediment remobilized, velocity of remobilized sediments, slope angle, and/or recurrence intervals. However, geology and hydrology are important factors in slope failure likelihood and tsunamigenic potential. Here physical experiments and natural observations demonstrate the importance of excess pore pressures and sedimentology (i.e., clay content) for assessing failure likelihood and

tsunamigenic potential. The implications of excess pore pressure and clay concentration in future risk assessments are characterized. Excess pore pressure is identified as capable of lowering slope resistance to failure, triggering non-tsunamigenic slope failure, and preconditioning a slope for tsunamigenic failure. Excess pore pressures within slopes reduce slope stability and thus areas with excess pore pressures should be considered at increased risk. Bathymetric features such as pockmarks/craters and surface eruptions/mud volcanism are identified as past or present indicators of excess pore pressures.

Clay concentration is identified as determining slope failure's potential for generation of intact blocks, a requisite for tsunamigenic land sliding. Measurement of slope clay concentrations are shown to provide a first-order estimation of failure behaviour to be tsunamigenic. Overall, this work notes that understanding how a slope might be preconditioned for failure is a necessary input for a more holistic risk assessment.

## 1 Introduction and Motivation

Subaqueous landslides can present a potential hazard (cause loss of life, injury or other health impacts, property damage, social and economic disruption, or environmental degradation; UNISDR, 2009; UNDRR 2016) to nearby communities through their ability to damage seafloor/lake-floor infrastructure (e.g., cables, pipelines, oil wells; Nadim and Locat, 2005; Fine et al., 2005; Zakeri, 2008; Vanneste et al., 2013) and generate tsunamis (McAdoo and Watts, 2004; McAdoo et al., 2004; Tappin et al., 2008; ten Brink et al., 2009, 2014; Harbitz et al., 2014b; Normandeau et al., 2019). Evidence for such failures has been found

globally (Lemke, 1967; Bondevik et al., 2005; Kvalstad et al., 2005; Tappin et al., 2008; Løvholt et al., 2019), including in



fjords (Lastras et al., 2013; Harbitz et al., 2014a; Paris et al., 2019) and lacustrine (Mountjoy et al., 2019; Kremer et al., 2021) environments.

Analysis of subaqueous landslides worldwide has documented that landslide initiation and tsunamigenesis often occur on low angle (< 2°) slopes (Prior et al., 1986; Watts, 1997; Haflidason et al., 2004; Bryn et al., 2005; Urlaub et al., 2012). Investigation
of tsunamigenic subaqueous landslides document that larger failures occur on passive margins in comparison to active margins (Kvalstad et al., 2005; Urlaub et al., 2012; Harbitz et al., 2014a; Løvholt et al., 2019; Paris et al., 2019). Events in either margin type, however, can be devastating, and efforts continue to increase preparedness and resilience (the ability of a system, community, or society exposed to hazards to resist, absorb, accommodate, adapt to, transform, and recover from the effects of a hazard in a timely and efficient manner, including through the preservation and restoration of its essential basic structures
and functions through risk management; UNISDR, 2009; UNDRR 2016) to such events (Amir et al., 2013; Goda et al., 2019; Matti et al., 2023). Currently these efforts focus on preparation and response (actions taken directly before, during, or immediately after a disaster in order to save lives, reduce health impacts, ensure public safety, and meet the basic subsistence needs of the people affected; UNISDR, 2009; UNDRR 2016) on active margins (Tappin et al., 2001; McAdoo et al., 2004; Amarasiri De Silva, 2009; Scholz et al., 2016; Imai et al., 2019; Goda et al., 2019), do not fully characterize passive margins
where failures and tsunamis are less frequent but larger (Kvalstad et al., 2005; Urlaub et al., 2012; Harbitz et al., 2014a; Løvholt et al., 2019; Paris et al., 2019), and do not address event likelihood and severity.

Previous studies have assessed risk (a qualitative or quantitative approach to determine the nature and extent of disaster risk by analyzing potential hazards and evaluating existing conditions of exposure and vulnerability that together could harm people, property, services, livelihoods, and the environment on which they depend; UNISDR, 2009; UNDRR 2016) to
landslide-generated tsunamis (Locat et al., 2010; Moscardelli et al., 2010a; Geist and Ten Brink, 2012; Amir et al., 2013; ten Brink et al., 2014; Goda et al., 2019; Matti et al., 2023). However, they primarily focus on vulnerability (the conditions determined by physical, social, economic, and environmental factors or processes which increase the susceptibility of an individual, a community, assets, or systems to the impact of hazards; UNISDR, 2009; UNDRR 2016) and overlook underlying processes and controls regarding the hazard (i.e., slope destabilization and potential tsunamigenesis). Underlying landslide
processes and controls are important to understanding the hazard presented by a slope failure, as such characteristics determine the tsunamigenic potential of the failure (Terzaghi et al., 1996; Sawyer et al., 2012; Sawyer and DeVore, 2015a; ten Brink et al., 2016; Silver and Dugan, 2023a). Inclusion of these features can improve risk assessments and associated forecasting of slope failure likelihood, likelihood of slope failure to be tsunamigenic, and identification of areas at risk of slope failure.

To improve understanding of subaqueous landslide and related tsunami risk, we identify hydrologic and geologic conditions
that are important for assessing the likelihood of initiation, the location of initiation, and the tsunamigenic potential of land sliding. The factors presented here have been largely overlooked when assessing subaqueous landslide risk, particularly as they relate to passive margins, such as the North American Atlantic coast (ten Brink et al., 2009, 2014), the Gulf of Mexico (ten Brink et al., 2006; Stigall and Dugan, 2010; Long et al., 2011), and the North Sea (Hühnerbach and Masson, 2004; Bondevik et al., 2005; Bryn et al., 2005; Solheim et al., 2005). We identify excess pore pressure (pore pressures in excess of



hydrostatic conditions) and clay content as factors that have important implications in risk assessment but have hitherto been omitted from risk assessment tools. We identify bathymetric and subsurface indicators of excess pore pressure and related seafloor deformation features which can be applied to natural environments as evidence of past or present excess pore pressures. We also characterize changes in slope behavior and land slide tsunamigenic potential based on clay content, a characteristic easy to measure in sediment samples.

Inclusion of excess pore pressure and clay content in risk assessments will improve accuracy in assessment of slope failure likelihood and tsunamigenic potential, improving accuracy of tsunamis severity assessment and identification of at-risk areas. We show that omission of excess pore pressures in calculations results in an overprediction of slope stability, increasing with clay content. Thus, consideration of pore pressure and clay content as shown here can correct calculations and models to better constrain slope stability conditions. We also show that these two factors can precondition a slope for tsunamigenic failure as

increased pore pressures decrease resistance to slope failure and increased clay content increases tsunamigenic potential of failures, as increased clay content favors blocky failure over fluidized failure behavior. This is of particular concern in passive margins, (e.g., the North Sea, Atlantic Ocean, and the Gulf of Mexico; Bryn et al., 2005; Fine et al., 2005; Stigall and Dugan, 2010; respectively) where tsunamigenic failures are less frequent (Sawyer and DeVore, 2015b), but are larger relative to active margins and thus can cause severe damage to communities not typically concerned with tsunamis (Fine et al., 2005; Schulten

et al., 2019).

Including excess pore pressure in assessments will improve assessment of slope failure likelihood by increasing quantification of current slope stability versus slope destabilization criteria. Including clay concentration in assessments will improve the forecasting of slope failure behavior and efforts to predict if failure will be tsunamigenic or non-tsunamigenic.

## 2 Background

Tsunamis can devastate communities (Lemke, 1967; Mcadoo et al., 2008; Tappin et al., 2008; Muhari et al., 2018; Løvholt et al., 2019; Paris et al., 2019) and can damage critical coastal infrastructure, such as nuclear power plants (McHugh et al., 2016), fisheries and related equipment (United Nations News, 2005), harbors, and utilities even with low tsunami wave-heights (US Department of Commerce; Tewfik et al., 2008; Garces et al., 2010; ten Brink et al., 2014; Tappin et al., 2014; Williams et al., 2022). The 2004 Sumatra-Andaman mega-earthquake (magnitude 9.2; Shearer and Bürgmann, 2010) was one of the worst

natural disasters in modern history (Fujii and Satake, 2007; Shearer and Bürgmann, 2010). The produced tsunami reached heights of 30 – 50 m in some regions (Aceh Province, Indonesia; Shearer and Bürgmann, 2010), arriving at shorelines within 2 hours of the initial earthquake (Fujii and Satake, 2007). The tsunami damaged communities in Indonesia, Sri Lanka, Thailand, the Maldives, Sumatra, and India (Fujii and Satake, 2007) ultimately claiming over 285,000 lives (Shearer and Bürgmann, 2010; World Bank, 2012; The Editors of Encyclopedia Britannica, 2019). Damage to fisheries and aquaculture sectors was

extensive (United Nations News, 2005). Sri Lanka lost over 7,500 fishermen, 80% of their coastal fishing vessels, and devastation of 80% of main fishing harbors (United Nations News, 2005). In Indonesia, 70% of the small-scale fishing fleet



was lost (United Nations News, 2005; Garces et al., 2010). In Aceh province, one of the most severely impacted areas of Indonesia, more than 4 million people depended on the activities of the regional fishing industry (Tewfik et al., 2008). Direct damage to the Indonesian fishing industry in Aceh province alone exceeded 102 million USD, with economic losses due to 100 disruption of fishing activity in the province estimated to be more than 410 million USD (Tewfik et al., 2008).

Damage and recovery efforts from disasters related to subaqueous landslides, such as tsunamis, flooding, and earthquakes, have been found to exacerbate inequities in effected communities (Fothergill et al., 1999; Cutter et al., 2003; Amarasiri De Silva, 2009). An investigation into recovery efforts in Sri Lanka after the 2004 Indian Ocean tsunami found race had become the deciding factor in humanitarian aid distribution post-tsunami in the district of Ampara (Amarasiri De Silva, 2009). 105 Amarasiri de Silva (2009) found ethno-political relationships pre-tsunami further intensified inequities in this process. Particularly, economically disadvantaged Muslim communities in Ampara disproportionately suffered compared to the more politically powerful Sinhalese and Tamil communities, in terms of mortality rate, morbidity, and damage to property (Amarasiri De Silva, 2009). Prior to the tsunami, the Muslim minority in Ampara lived along the coast in segregated, ribbon-like discrete communities, heightening their vulnerability to the tsunami (Amarasiri De Silva, 2009). Furthermore, during 110 recovery efforts, the Muslim fishing community was shown to have been selectively deprived of the benefits of humanitarian aid (Amarasiri De Silva, 2009).

Events such as the 2004 Sumatra-Andaman tsunami, their immediate damaging effects, and inequities in impacts and recovery efforts, reinforce the need to better understand tsunami risk. Such impacts show that proper hazard assessment, preparation, and recovery is important for combating social inequities. Particularly, improved awareness of tsunami risk for potentially 115 affected communities is necessary, including communities along passive margins, where failures can be larger than on active margins (Kvalstad et al., 2005; Urlaub et al., 2012; Harbitz et al., 2014a; Løvholt et al., 2019; Paris et al., 2019). Here we present findings that, when included in hazard and risk assessments, will improve the accuracy of identification of areas at risk by quantifying the likelihood of a given section of slope to fail and for that failure to be tsunamigenic.

When considering tsunami risk, communities along active margins, such as the Cascadian, Alaskan, and Japanese margins, are 120 typically considered to be at higher risk than passive margin communities (Lemke, 1967; Von Huene and Lallemand, 1990; Bardet et al., 2003; Finn, 2003; Yalçıner et al., 2003; McAdoo et al., 2004; Bondevik et al., 2005; Fine et al., 2005; Bernard et al., 2006; Hornbach et al., 2007; Mcadoo et al., 2008; Tappin et al., 2008; Moscardelli et al., 2010b; Garces et al., 2010; Loveless and Meade, 2010; Harbitz et al., 2014a; Scholz et al., 2016; Fujiwara et al., 2017; Strasser et al., 2019; Schulten et al., 2019; Imai et al., 2019; Goda et al., 2019; Sun and Leslie, 2020; Ikehara et al., 2021; Matti et al., 2023). However, 125 tsunamigenic landslides occur along passive margins, and these failures are generally larger than on active margins and can generate large tsunamis (Kvalstad et al., 2005; Urlaub et al., 2012; Harbitz et al., 2014a; Løvholt et al., 2019; Paris et al., 2019). As there is a direct relationship between landslide volume and tsunami height (Murty, 2003), events in either margin types can be devastating (Amir et al., 2013; Goda et al., 2019; Matti et al., 2023). However, efforts to increase community resilience to such events along active and passive margins typically overlook forecasting event likelihood and severity, focusing on





preparation and response (Cutter et al., 2003; McAdoo et al., 2004; Amir et al., 2013; Normandeau et al., 2019; Mountjoy et al., 2019; Goda et al., 2019; Howley, 2021; Matti et al., 2023).

Forecasting slope failure is complex and difficult (McAdoo et al., 2004; Hustoft et al., 2009; Stigall and Dugan, 2010; Harbitz et al., 2014a; Thena and Mohan, 2016; Scholz et al., 2016; Pecher et al., 2018; Imai et al., 2019; Silver and Dugan, 2020, 2023a; Ikehara et al., 2021; Vick and Dugan, 2021). Arguments have been made for increased slope stability along seismically

active margins relative to passive margins due to repeated, low-energy seismic events (Sawyer and DeVore, 2015a). However, tsunamigenic subaqueous landslide events continue to occur on seismically active margins (i.e., Alaska, 1964 and Papua New Guinea, 1998; Lemke, 1967; Tappin et al., 2001, 2008). Furthermore, evidence has been found of ongoing, low-velocity remobilization of sediments off the seismically active New Zealand coast (Pecher et al., 2018; Vick and Dugan, 2021). This indicates that even with repeated, low-energy seismic events on active margins, tsunamigenic landslides are still a hazard along

these margins.

When considering hazard of tsunamigenic landslides, investigation of mechanics and preconditioning factors of subaqueous landslides has revealed the importance of excess pore pressure (pore fluid pressures in excess of hydrostatic conditions) in landslides (Terzaghi, 1956; Westbrook and Smith, 1983; Dugan and Flemings, 2000; Strout and Tjelta, 2005; Dugan, 2008; Hustoft et al., 2009; Stigall and Dugan, 2010; Dugan and Sheahan, 2012; ten Brink et al., 2014; Ikari and Kopf, 2015; Dugan

and Zhao, 2018; Silver and Dugan, 2020, 2023a; Vick and Dugan, 2021). Excess pore pressures have been found to lower slope resistance to failure (Terzaghi, 1956; Terzaghi et al., 1996)(Terzaghi, 1956; Terzaghi et al., 1996) and generate non-tsunamigenic slope failures, such as mud volcanism and slope creep (Silver and Dugan, 2020, 2023a), indicating that slopes with excess pore pressures have the potential to damage sea or lake-floor infrastructure. However, modelling efforts of tsunamigenic landslide evolution and risk assessment tend to overlook causal mechanics such as excess pore pressures,

focusing either on geometrical parameters (i.e., slope angle, volume, velocity, and runout distance) or on recurrence intervals (based on identified and dated landslide deposits; Driscoll et al., 2000; McAdoo and Watts, 2004; ten Brink et al., 2006; Moscardelli et al., 2010a; Strozyk et al., 2010; Urgeles and Camerlenghi, 2013; Harbitz et al., 2014b; Sun and Leslie, 2020). Physical experiments (e.g., Silver and Dugan, 2020, 2023a) have revealed a correlation between content of cohesive clay in slope deposits and failure behavior, including likelihood of failure to be tsunamigenic. Silver and Dugan (2023) show that clay

content affects likelihood of slopes to experience repeat failures and that systems with excess pore pressure and cohesive clay increase the potential for tsunamigenic landslides (Silver and Dugan, 2023a). However, excess pore pressures were found to be incapable of generating tsunamigenic landslides without additional slope-failure triggering factors (Silver and Dugan, 2023a).

Modern efforts to quantify tsunami hazard tend to further overlook local geology or hydrology and assume seismic trigger

sources (Abe, 1989; Papadopoulos and Imamura, 2001; Amir et al., 2013). When proposing a new tsunami severity scale, Amir et al. (2013) acknowledged the potential for local geology and hydrology to confound assessments via potential seismic attenuation or ground liquefaction, yet geology and hydrology were not considered in their assessment scale. Furthermore, Amir et al. (2013) and other intensity scales have assumed seismic activity is required to trigger tsunamigenic land sliding



(Abe, 1989; Strasser et al., 2017) and only high magnitude seismic events (magnitude ≥ 6.5) were considered for
tsunamigenesis forecasting (Amir et al., 2013). Additionally, other seismic factors, like distance to earthquake epicenter, are
rarely considered (Abe, 1989) when addressing tsunami severity. This is despite evidence suggesting earthquake magnitudes
as low as 5.0 are sufficient for tsunamigenic subaqueous landslide initiation in the Gulf of Mexico (ten Brink et al., 2009;
Stigall and Dugan, 2010) and magnitude ($M_w$) 5 earthquakes have been observed in the North Sea (Bryn et al., 2005). However,
Bryn et al. (2005) note that earthquake magnitudes greater than 5.0 may have been required to trigger the Storegga Slide in the
North Sea.

When attempting to assess subaqueous landslide tsunamigenic risk, recurrence-interval based statistical approaches are
typically used (Harbitz et al., 2014b). These techniques use topography, bathymetry, and geochemical age constraints to
connect onshore tsunami deposits with subaqueous slope failure deposits and assign dates to the interpreted events. Then, a
frequency or recurrence model is developed based on the time intervals between these events. Said model is then extended for
forecasting potential future slope failures. The produced intervals provide an average timeline of slope failures and temporally
coinciding tsunamis but offer no predictive capabilities beyond average frequency of failure events for a given region. They
tend to overlook slope failure casual mechanics, underlying slope failure processes, and failure-initiation controlling factors.
Factors such as these (e.g., pore pressures, clay content) can improve models by providing information on where failure will
occur and the likely behavior of the anticipated failure. Overlooking an underlying process such as slope failure rate can further
confound recurrence interval models, as slow-moving, non-tsunamigenic slope failure deposits will be included in recurrence
models with faster, tsunamigenic failures. Furthermore, these models often ignore the composition of slope failure deposits
(e.g., clay content and relative cohesion), causing the models to assume all deposits and their related slope failures were
tsunamigenic, which we show here to be a false assumption. Therefore, including such factors would improve understanding
of the investigated system's current stability and identification of areas at highest risk of future failures. Finally, these models
also often ignore or assume specific triggering mechanisms, overlook effects from sea-level change, and effects of local
geology and hydrology (Harbitz et al., 2014b), in part due to limitations of available data.

Here, we identify key hydrologic (pore pressure) and geologic (clay content and cohesion) conditions that need to be considered
in future tsunamis risk assessments based on physical experiments and investigations of past tsunamigenic subaqueous
landslides. These conditions are excess pore pressures and cohesive clay concentration. Integration of these parameters in
assessments will improve tsunami severity assessment and at-risk area identification, particularly in overlooked passive
margins, by improving characterization of slope failure likelihood and tsunamigenesis likelihood.

## 3 Excess Pore Pressures – Decreasing Resistance to Flow

Physical experiments that replicate subaqueous landslides reveal the importance of excess pore pressures in destabilizing
slopes and preparing slopes for tsunamigenic failures (Silver and Dugan, 2023a). These experimental results are generally
consistent with slope failure models (Terzaghi, 1956; Dugan and Flemings, 2000; Nixon and Grozic, 2007; Stigall and Dugan,





2010; Ikari and Kopf, 2015). The physical experiments showed that excess pore pressures can cause slope failures, but alone cannot trigger tsunamigenic failures as blocky failures were produced but did not migrate downslope (Silver and Dugan, 2023a). Extending on the process analyses, Silver and Dugan (2023) compared excess pore pressures required to induce failure to numerical models used to predict slope stability and found commonly used models overpredict excess pressure required for

failure in most cases, particularly for clay concentrations common to the marine environment (e.g., ≥ 25% clay by wt.; **Fig. 1**). Thus, existing models may suggest environments are more stable than they really are, and smaller perturbations, such as small magnitude earthquakes, could create tsunamigenic slides when excess pore pressures are present, especially in clay-dominated margins.

Excess pore pressure reduces slope stability by counteracting normal stresses exerted on the slope by overlying sediments

which reduces the frictional resistance and favors landslide movement (Freeze and Cherry, 1979; Terzaghi et al., 1996). Generation of excess pore pressure in marine clays and the resulting slope destabilization caused by rapid sediment loading has been proposed as a contributing factor to the 8200 B.C.E. Storegga Slide Slide (Haflidason et al., 2005; Kvalstad et al., 2005; Bryn et al., 2005), the largest identified subaqueous landslide deposit. The tsunami generated by the event deposited sediments over 20 m high and as far away as Scotland (Bondevik et al., 2005).


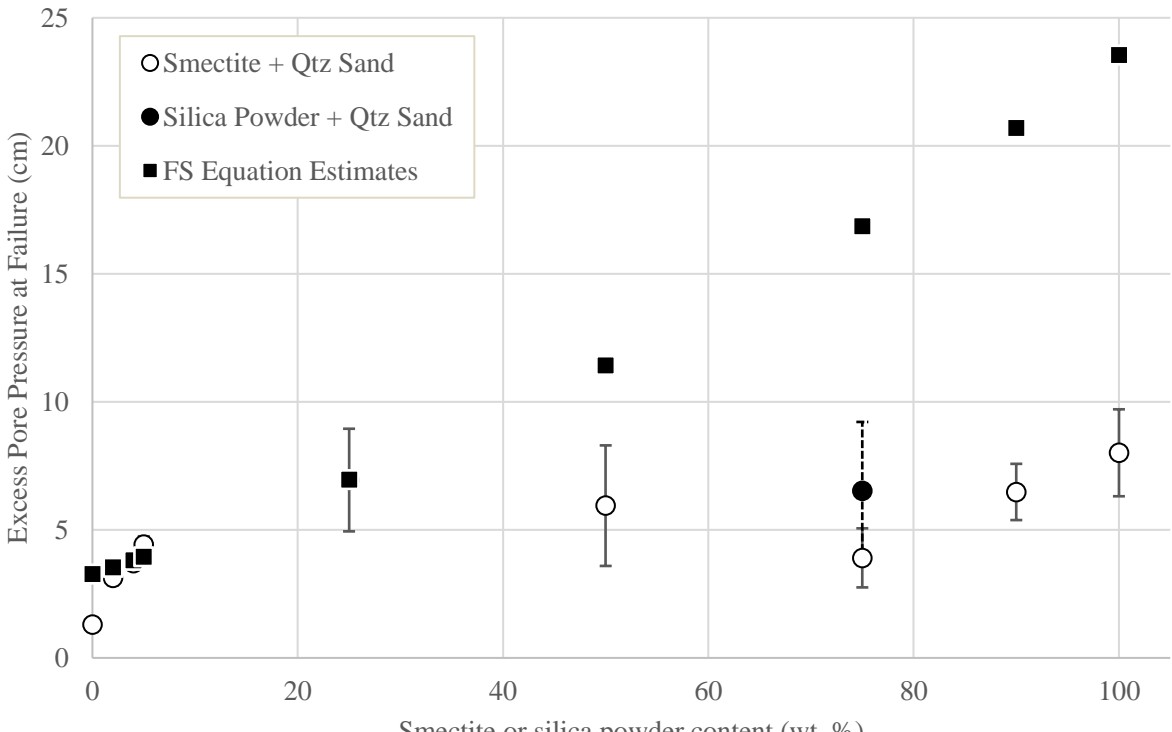

**Figure 1. A comparison of excess pore pressure required to induce slope failure predicted by the infinite slope Factor of Safety (FS) equation and critical excess pore pressure measured in experiments as a function of smectite clay (cohesive) or silica powder (non-cohesive, clay sized grains) content. Modified for clarity from Silver and Dugan (2023).**





Figure 2. A comparison of (A) a cross-section of the lower part of the Storegga Slide from Bryn et al., (2005) depicting fluid flow caused by excess pore pressure in the subsurface and (B) fluid flow, cavity formation, and surface eruption caused by excess pore pressure in physical experiments performed by Silver and Dugan (2023). Figure 2 (A) is redrawn from Bryn et al., (2005).






In an analysis of the Storegga slide, Bryn et al. (2005) interpreted that excess pore pressure in the subsurface was crucial to
the large-scale failure, qualitatively matching phenomena observed in physical experiments (Silver and Dugan, 2023a; **Fig. 2**).
In experiments, generation of connected subsurface fractures/fissures by excess pore pressure created preferential flow
pathways for fluid movement and generated weak layers (Silver and Dugan, 2023a; **Fig. 2**). In the natural environment, these
weak layers act as preferential locations for sediment shearing and could increase the likelihood for blocky slope failure, a
necessity for a landslide to be tsunamigenic.

The identification of excess pore pressures in subaqueous slopes can thus improve slope hazard assessments by 1) identifying
specific locations and/or sediment layers within a slope with reduced slope stability, 2) improving quantification of slope
stability conditions past, present, and future, and 3) identifying areas where further surveying (i.e., seismic or EM) is necessary
to investigate the potential presence of rafted blocks, a feature that increases the likelihood for failure to be tsunamigenic.

## 4 Clay Content – Failure Shape and Size

In the presence of excess pore pressure, cohesive clay concentration was observed to change slope behavior in physical
experiments representative of marine delta, coastal, and shelf environments (Silver and Dugan, 2020, 2023a). At low clay
concentrations (< 25% wt.), slope failures occurred as local surface eruptions, sediment excavation, and deposition (e.g., mud
volcanism; **Figs. 3A, 4C, and 5C**). Surface eruptions produced conical sediment deposits and created high permeability flow
pathways that prevented further accumulation of excess pore pressure, thus mitigating subsequent failures. In these sand-rich
environments, no evidence for tsunamigenic slope failure preconditioning was found. With higher concentrations of clay
(common in the marine environment; 25 – 90% wt clay), separation of intact sediment blocks from the parent slope were
produced (rafting; **Fig. 6**). However, these rafted blocks did not remobilize, which is required for tsunamigenesis. At clay
concentrations ≥ 25% (wt.), slopes experienced multiple failure events when excess pore pressure was present, without
additional increases in excess pore pressure. These findings indicate that 1) in sand-rich environments (5% clay or less), once
surface eruption occurs, no additional failure features should develop, 2) sand-rich environments are less likely to experience
tsunamigenic subaqueous slope failures than slopes with higher clay concentrations (≥ 25% wt.) due to the lack of rafted block
development, 3) environments with ≥ 25% wt. clays are more likely to experience multiple failure events than environments
with less clays, and 4) in environments with ≥ 25% wt. clays, a surficial slope failure (e.g., surface eruptions) does not indicate
a further stabilizing of the slope, lowering of excess pore pressures, or a lowered likelihood for additional slope failures.

The quantification of clay content in subaqueous slopes can thus improve slope hazard assessments by providing a first order
forecast of slope failure behavior relevant to tsunamigenic potential (e.g., surface eruptions under low clay concentrations vs.
weak layer or rafted block formation with clay concentrations ≥ 25% wt.).





## 5 Applications to Natural Settings

Observations of excess pore pressure-induced phenomena provide

clues for identification of excess pore pressure in the natural environment. In physical experiments, excess pore pressure consistently produced surface doming/swelling of sediments, tension cracking, sediment/mud volcanism (eruptions), and sediment remobilization deposits, all surficial processes that could be observed

via bathymetry without measuring subsurface processes (**Figs. 2 and 3**). In the case of sediment/mud volcanism, craters or pockmarks developed in physical experiments. Pockmarks are a widespread depositional phenomenon in the marine environment (King and MacLean, 1970; Hovland and Judd, 1988; Hovland, 2003) and suggest

previous and/or ongoing expulsion of fluids/gas (Yun et al., 1999). In experiments, Silver and Dugan (2023) produced sediment/mud volcanism with deposits resembling pockmarks identified offshore Norway (Hovland, 2003) and California (**Fig. 5;** Yun et al., 1999). In the case of California, the identified pockmarks and associated excess

pore pressure have been suggested to have contributed to a large mass-wasting event (Humboldt Slide; Yun et al., 1999). Upon further investigation of excess pore pressure and associated phenomena, experimentally observed eruption mechanisms matched proposed mechanisms for natural eruptions based on field observations (**Fig. 5;**

Yun et al., 1999; Dimitrov, 2002; León et al., 2007; Silver and Dugan, 2023).

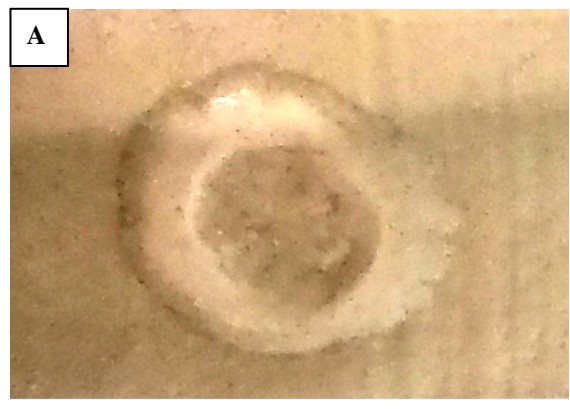

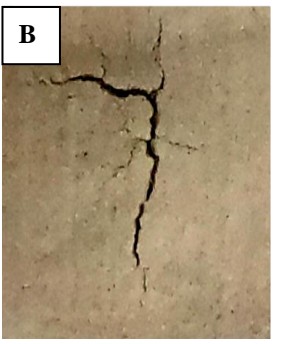

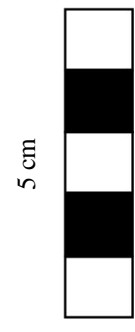

**Figure 3.** Top-down images of flume experiments of submerged, saturated, subaqueous slope failures. No vertical or horizontal exaggeration. **(A)** Surface expression of subsurface excess pore pressures in the form of a conical sediment remobilization deposit as the aftermath of sediment/mud volcanism in 2% wt. cohesive clay + 98% wt. non-cohesive sand mixtures. **(B)** Surface expression of subsurface excess pore pressures in the form of surface swelling and tension cracking.

Therefore, surficial seafloor bathymetric features identified in the natural environment such as pockmarks, swelling, and non-meteoric craters may be expressions of subsurface excess pore pressure and can be used to identify past and/or present excess pore pressure.

Silver and Dugan (2023) also demonstrated that excess pore pressure in subaqueous slopes with clay concentrations consistent with marine environments ($\geq 25\%$ wt.) could precondition a slope for tsunamigenic failure via the formation of rafted sediment blocks. Seismic energy interacting with these blocks with sufficient force may mobilize the blocks downslope, potentially generating a tsunami, consistent with previous work on passive margin failures in the North Sea (Storegga slide; Kvalstad et al., 2005; Bryn et al., 2005) and in the Gulf of Mexico (Ursa slide; Stigall and Dugan, 2010). Understanding block formation

is critical as the rapid mobilization of intact sediment blocks is required for tsunamigenesis. Furthermore, the boundaries



**Figure 4.** Images of pockmarks in nature and experiments. (A) A shaded high-resolution multi-beam bathymetry image offshore Norway showing pockmarks and a gas-transport pipeline. Reproduced from Hovland (2003). (B) Pockmarks identified near the headscarp of the Humboldt Slide offshore California as imaged by multibeam high-resolution bathymetry. Reproduced from Yun et al., (1999). (C) Pockmarks developed in a physical flume experiment of a sand-rich subaqueous slope performed by Silver and Dugan (2020). Pockmark features developed during the experiment visually resemble pockmarks in the natural environment, such as those in A and B, and were formed by excess fluid/pore pressures, matching proposed formation mechanisms for the natural environment.

between rafted blocks and the parent slope acts as preferential shear planes which decrease the amount of seismic energy required to mobilize the block. Thus, the magnitude of seismic event required to mobilize the block is lower than if it was not separated from the parent slope.

It is important to identify excess pore pressure and possible rafted sediment blocks when assessing tsunami risk for a coastal/lacustrine community. Should a risk assessment overlook the presence of excess pore pressure, subsurface fractures, and/or rafted sediment blocks, it may overestimate the seismic magnitude required to trigger a tsunamigenic subaqueous landslide. Excess pore pressure has been found in passive margins such as the Atlantic margin offshore



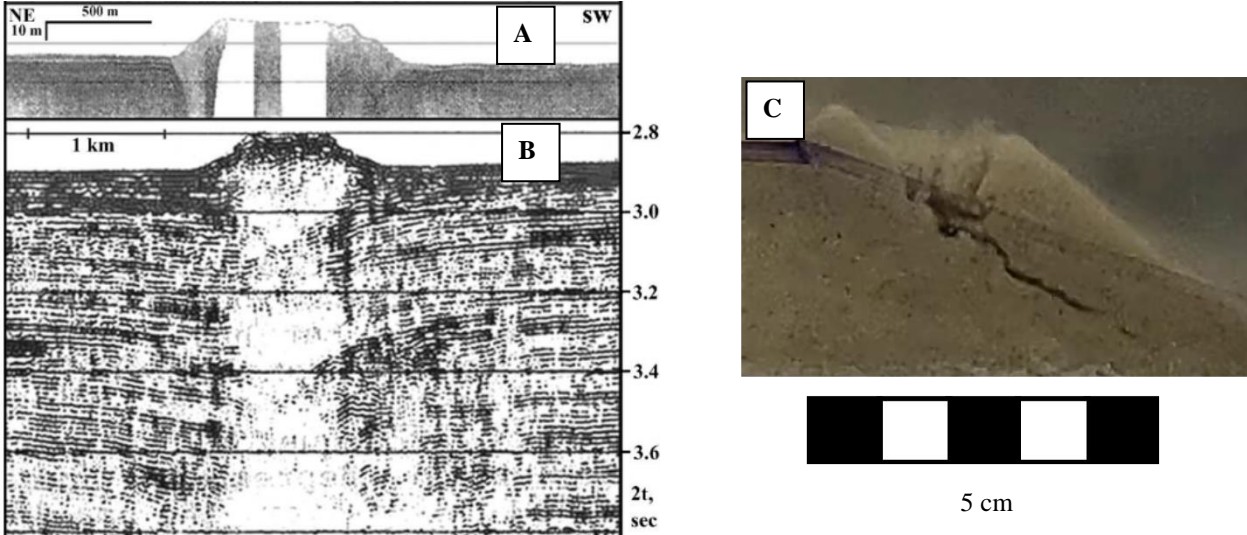

**Figure 5. A deep tow sub-bottom profilograph line (A) and seismic section (B) depicting a cross-section of the Malishev mud volcano in the Black Sea. Reproduced from Dimitrov (2002). (C) A cross-section of a mud volcano developed in a physical flume experiment of a sand-rich subaqueous slope performed by Silver and Dugan, (2020). The mud volcano developed during the physical experiment visually resembles mud volcano deposits found in nature, such as the one depicted in A and B, and was formed by excess fluid/pore pressures, matching proposed formation mechanisms for the natural environment.**

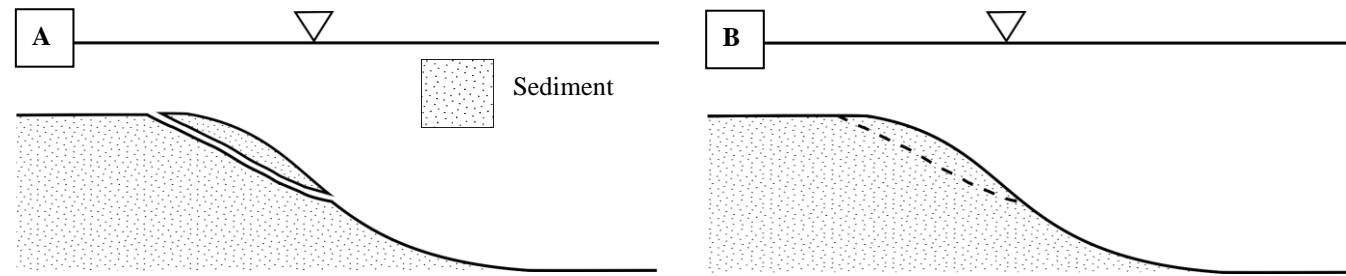


**Figure 6. A schematic of slope preconditioning caused by excess pore pressures. An intact block of sediment is separated and rafted from the parent slope when excess pore pressure is exerted. Additional forces, perhaps seismic, are needed to mobilize the block downslope, potentially generating a tsunami. The triangles in (A) and (B) indicate sea level. The dashed line in (A) represents a subsurface crack, fissure, and/or void formed by excess pore pressure. In step (B), this crack/fissure/void has connected with other**
**such features to separate a block of sediment from the parent slope. Based on findings from Silver and Dugan (2023).**

New Jersey (Dugan and Flemings, 2000), the Gulf of Mexico (Stigall and Dugan, 2010; Long et al., 2011), and the Norwegian margin of the North Sea (Bondevik et al., 2005; Solheim et al., 2005). These margins also have clay concentrations high enough (≥ 25% wt.; Dugan and Flemings, 2000; Forsberg and Locat, 2005; Sawyer et al., 2008) for the potential formation of
rafted sediment blocks and repeat failures. Should an earthquake occur with an epicenter and/or sufficient strength in these regions, a tsunamigenic landslide may be triggered (Stigall and Dugan, 2010). Therefore, risk managers for these regions and



other passive margins need to consider the local geology (clay concentration, rafted blocks) and hydrology (excess pore pressure) in tsunami risk assessments.

## 6 Conclusions

Subaqueous landslides present a hazard to communities through their ability to damage infrastructure and generate tsunamis. Such events and their consequences have been shown to exacerbate inequities in effected communities. Current efforts to assess tsunamis risk tend to depend on failure geometric properties or failure recurrence intervals. However, comparison of physical experiments and nature observations identify excess pore pressure and clay concentration are important factors in slope failure likelihood and tsunamigenic potential. Excess pore pressure can precondition a slope for tsunamigenic failure

through slope destabilization and formation of rafted sediment blocks when clay concentrations are 25 – 90% wt., which is common in the marine environment. Identification of excess pore pressure in natural marine environments will improve tsunami risk assessment, by identifying areas with reduced slope stability and when considering magnitude of seismic event needed to remobilize sediments. Bathymetric features such as non-meteoric craters or pockmarks, and surface eruptions/mud volcanism can all be used as possible identifiers of past or current excess pore pressures. Characterization of clay content and

identification of rafted sediment blocks will improve tsunami risk assessment by improving forecasting of failure behavior and potential for sediment remobilization to be tsunamigenic.

## 7 Acknowledgements

This work was funded by NSF-OCE-1753680 and the Colorado School of Mines.

## 8 Competing Interests

The contact author has declared that none of the authors have any competing interests.

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
