# Peer review of "Improving Risk Assessment of Subaqueous Landslides and Tsunami Generation by Understanding the Roles of Pore Pressure and Sediment Cohesion"

_EGUsphere, 2024_

## Author Comment (AC2)

EGUSphere Preprint Responses: June 2024

CC1:
*This manuscript conducts the risk analysis of subaqueous landslides and tsunami. Generally, this manuscript is not well organized and the novelty is very limited. As a result, a rejection is recommended.*

1. *The Introduction section requires to be reorganized. The current version looks a little chaotic. The current research status should be organized in a more logistic way and then the research potentials should be outlined clearly.*

   Response: Thank you for the helpful feedback. We will happily combine and shorten the introduction and background/literature review sections into a single section of 2-3 pages, which will improve the readability of the manuscript.

2. *The Background section is too superfluous. Simplification is recommended.*

   Response: our revisions to the introduction and background (see response 1 above) will streamline and simplify the background.

3. *For the results in Fig. 1: Please explain how the excess pore water pressure to trigger slope failure is calculated.*

   Response: Thank you for identifying this as an area of confusion in the manuscript. We will revise the manuscript to be clearer. Excess heads required to trigger slope failure ($h^*$) were measured directly during experiments by monitoring and documenting head throughout experiments including at the time of failure. The moment of failure initiation was identified in video recordings. We will revise the manuscript to detail these methods and experiment findings. We will also convert excess head to unitless normalized overpressure ($\lambda^*$):

$$\lambda^* = \frac{\rho_w g h^*}{(\rho_b - \rho_w) g h}$$

   where $\rho_w$ is water density, $g$ is gravitational acceleration, $\rho_b$ is bulk density, and $h$ is the depth from the water-sediment interface to the sediment-cobble interface. This normalized overpressure includes data on total stress, hydrostatic pressure ($\rho_w gh$), and overpressure ($\rho_w gh^*$), making it scalable and more relevant to global applications in natural submarine slopes. This will be detailed and included in the revised manuscript, as well as relevance to hazard assessment.

4. *Fig 1: Smectite and Silica Powder are used for comparison. However, the authors seem not to describe the differences between these two cases.*

Response: we will expand our explanation of why we chose quartz, smectite, and silica powder and use this to improve our discussion on the importance of these two investigations, their differences, and their impacts.

5. *As the title indicated, the risk analysis should be the focus of this study. As for risk analysis, the estimation index should be failure probability, reliability index, sliding volume, etc. However, in the current version, such contents are all missing.*

   Response: Thank you for helping us understand how we can improve clarity of the study's focus and utility. It was our intent to provide two key findings: the first to demonstrate that the accounting of excess pore pressures in simple, effective numerical models for slope stability (e.g., the Factor of Safety equation) is unreliable and importantly over-predicts slope stability conditions under clay concentrations relevant to natural marine environments, and 2) that clay concentration can be used as a first-order estimate of slope failure behavior for hazard assessment. It is not our intention to reorganize or revise specific failure probability indices, but rather to provide an important geotechnical perspective that the hazard assessment community can incorporate into their assessments. We will therefore change the title of the paper to better reflect these aims. We propose instead: "The importance of clay and pore pressure in submarine slope failures; implications for forecasting."

---

## Author Comment (AC3)

EGUSphere Preprint Responses: June 2024

RC1:

*I would like to thank authors to submit this manuscript to the Natural Hazards and Earth System Sciences, it could be considered to publish after addressing the following comments.*

*General comment:*

*-The study explores the intriguing subject of "the role of hydrogeology and geology in assessing tsunami potential." However, the paper's structure could benefit from adjustments regarding the volume and distribution of content across sections. Currently, the layout resembles a review paper, but it is not! Five pages dedicated to the introduction and literature review: three pages on background information and two pages for the introduction within a 13-page paper. If this paper is not intended to be a review paper, the authors should combine the introduction and literature review into one section with two to three pages.*

> Response: Thank you for the helpful feedback. We will happily combine and shorten the introduction and background/literature review sections into a single section of 2-3 pages, which will improve the readability of the manuscript.

*- Another concern is the frequent referencing of previous studies by the authors or co-authors, which detracts from the paper's independence. I believe each paper must have a certain level of independency; I am not seeing the satisfactory level of this concern in this paper. E.G., Physical experiment that were done, why there is not enough details of experiments were not stated here!?*

> Response: This is very useful guidance. We will revise the manuscript to include more details about the experiments while maintaining our intent to give overviews of the experiments that will be engaging to those specifically interested in issues related to hazard/risk while at the same time providing new insights for all readers of EGUSphere. Specifically: we will include more details from the experiments as they are pertinent to our discussion points: 1) general overview of experiment setup, including sediment mixtures and relevance to natural settings, scale, the number of experiments, slope stabilizing forces, and slope destabilizing forces, 2) trends in experiment results with sediment mixtures, and 3) trends in experimental results with pore pressures required for slope failure initiation.

*Specific comments:*

*- " However, comparison of physical experiments and nature observations identify excess pore pressure and clay concentration are important factors in slope failure likelihood and tsunamigenic potential." Given the qualitative nature of the content, it is unclear why it has been reported statistically. Could you clarify the significance of these statistics and their potential relevance to the study?*

Response: Guidance on clarifying communication of our data at this stage is very helpful indeed Our thoughts were that the ranges of clay concentrations presented here can provide a framework for hazard assessment to compare against, i.e., hazard assessors can measure the clay concentration of a submarine slope and compare to our findings, providing a first-order estimate of their slope's likely failure behavior.

Our intent with our pore pressure findings was to communicate demonstrated inaccuracies in slope stability modelling and importantly, over prediction of slope stability using a simple numerical model (the Factor of Safety equation).

We will revise the manuscript to include a table with the overpressures required to induce slope failure as a function of clay content as well as the equivalent normalized overpressure, which allows scaling the experimentally observed overpressure and effective stress conditions at failure to natural systems. We will also include overpressure measurements from literature related to natural submarine slope failure, but these are limited. We will further communicate how this motivates further measurement of in-situ submarine slope pore pressures.

*-The incorporation of laboratory shear strength tests on sediment samples from various depths and locations within the study area could provide empirical data to better correlate cohesion with the initiation and propagation of submarine landslides. This should be considered if it is possible.*

Response: Thank you for the guidance. Indeed, the inclusion of such data would create a more robust correlation with cohesion and failure initiation. We understand that the manuscript in its current form creates an expectation to be presented with such data. However, the intent of this paper is not to present a specific study area, but a general physics- and mechanics-based investigation of submarine slope destabilization which can be applied globally. Collection of sediment samples and characterization of their stability-relevant characteristics (e.g., shear strength, pore pressure, consolidation history, etc.) is beyond the scope of this paper, in our opinions. We will revise the manuscript to better clarify its global generalized applicability.

*Technical corrections:*

*I have noticed some typos, such as the ones below. I recommend reviewing the paper more carefully to avoid these errors.*

Response: We will do a careful and thorough copy edit of our manuscript to rectify these errors. Thank you for bringing them to our attention.

*"Measurement of slope clay concentrations are (is) shown to provide a first-order estimation of failure behaviour to be tsunamigenic."*

*" to investigate the potential presence of rafted blocks, a feature that increases the likelihood for (of) failure to be tsunamigenic."*